# Imaging Approach in the Diagnostics and Evaluation of the Psoriasis Plaque: A Preliminary Study and Literature Review

**DOI:** 10.3390/diagnostics14100969

**Published:** 2024-05-07

**Authors:** Mircea Negrutiu, Sorina Danescu, Theodor Popa, Monica Focsan, Stefan Cristian Vesa, Florin Szasz, Adrian Baican

**Affiliations:** 1Department of Dermatology, “Iuliu Hatieganu” University of Medicine and Pharmacy, 400012 Cluj-Napoca, Romania; negrutiu.mircea.ionut@elearn.umfcluj.ro (M.N.); sorina.danescu@umfcluj.ro (S.D.); abaican@umfcluj.ro (A.B.); 2Department of Rehabilitation, “Iuliu Hatieganu” University of Medicine and Pharmacy, 400012 Cluj-Napoca, Romania; popa.theo94@gmail.com; 3Nanobiophotonics and Laser Microspectroscopy Center, Interdisciplinary Research Institute on Bio-Nano-Sciences, Babes-Bolyai University, 400271 Cluj-Napoca, Romania; monica.iosin@ubbcluj.ro; 4Department of Functional Sciences, Discipline of Pharmacology, Toxicology and Clinical Pharmacology, Faculty of Medicine, “Iuliu Haţieganu” University of Medicine and Pharmacy, 400012 Cluj-Napoca, Romania; stefanvesa@gmail.com; 5Department of Obstetrics and Gynaecology, Faculty of Medicine and Pharmacy, University of Oradea, 410073 Oradea, Romania

**Keywords:** psoriasis plaque, dermatoscopy, videodermatoscopy, ultrasonography, confocal microscopy

## Abstract

(1) Background: the aim of the study was to demonstrate its usefulness in the field of imaging evaluation of plaque morphology in psoriasis vulgaris, with an emphasis on the use of confocal microscopy and other advanced skin-imaging techniques. (2) Methods: we conducted a prospective study over two years (July 2022–April 2024), on patients diagnosed with moderate or severe psoriasis vulgaris, treated in the dermatology department of our institution. We selected 30 patients, of whom 15 became eligible according to the inclusion and the exclusion criteria. A total of 60 psoriasis plaques were analyzed by dermatoscopy using a Delta 30 dermatoscope and Vidix 4.0 videodermoscope (VD), by cutaneous ultrasound (US) using a high-resolution 20 MHz linear probe, and by confocal microscopy, along with histopathological analysis. (3) Results: the study included fifteen patients with vulgar psoriasis, diagnosed histopathologically, of whom six were women and nine were men, with an average age of 55. Between two and six plaques per patient were selected and a total of sixty psoriasis plaques were analyzed by non-invasive imaging techniques. Twelve lesions were analyzed with ex vivo fluorescence confocal microscopy (FCM), compared to histology. US showed that the hyperechoic band and the lack of damage to the subcutaneous tissue were the most common criteria. The epidermis and dermis were found to be thicker in the area of psoriasis plaques compared to healthy skin. Dermatoscopy showed that the specific aspect of psoriasis plaques localized on the limbs and trunk was a lesion with an erythematous background, with dotted vessels with regular distribution on the surface and covered by white scales with diffuse distribution. The presence of bushy vessels with medium condensation was the most frequently identified pattern on VD. Good correlations were identified between the histological criteria and those obtained through confocal microscopy. (4) Conclusions: the assessment and monitoring of patients with psoriasis vulgaris can be conducted in a more complete and all-encompassing manner by incorporating dermatoscopy, ultrasonography, and confocal microscopy in clinical practice.

## 1. Introduction

Psoriasis is considered to be the immune-mediated disease with the highest prevalence in the world (2–5% of the world’s population) [1]. According to the National Institutes of Health, 7.5% of the American population suffers from psoriasis. The prevalence of psoriasis varies by nation and age, indicating that ethnicity, genetic background, and environmental factors influence the onset of psoriasis [1,2].

The disease’s etiology has been attributed to a combination of extrinsic and intrinsic factors, including mechanical stress, air pollution, sun exposure, medications, vaccinations, infections, and lifestyle [1]. The human leukocyte antigen HLA-Cw6 is the allele that poses the greatest risk for psoriasis [3]. Clinically, according to the International Psoriasis Council, four forms are described: in plaques, guttate, generalized pustular psoriasis, and erythrodermic. Regarding the incidence of the disease according to age, a dual model is described, with two peaks of onset, one around the age of 30–39 years and the second peak around the age of 60–69 years [4].

The most common type of psoriasis encountered is plaque psoriasis, with a proportion of 85–90%. It is characterized by oval or irregular, erythematous plaques covered with white scales. They are located especially in the extensor areas, the elbows, knees, scalp, and lumbar area, but they can affect any part of the body, often with symmetrical distribution [5].

Psoriasis is a dynamic pathology, so in the initial stages the first change appears in the highest layer of the dermis (papillary dermis). The blood vessels become dilated, and tortuous with ectasized lymphocytes and neutrophils migrating towards the epidermis. Subsequently, the proliferation and migration of aberrant keratinocytes occur, resulting in the thickening of the epidermis, the loss of the granular layer, and the appearance of parakeratosis. In the advanced stages, psoriatic hyperplasia appears, characterized by acanthosis (thickening of the spinous layer), papillomatosis, and elongation of rete ridges [6].

Psoriatic arthritis, metabolic syndrome, depression, non-alcoholic fatty liver disease, Crohn’s disease, lymphoma, and cardiovascular issues are among the chronic illnesses that people with psoriasis are more likely to develop [7]. In order to prevent these complications, a correct diagnosis is necessary as early and as accurately as possible. The diagnosis of psoriasis is mostly clinical, and in less obvious cases the histopathological examination is used. The pathological changes in the psoriasis plaque depend on the age of the lesion. Aspects such as dilated capillaries in the papillary dermis, lymphocytic inflammatory infiltrate, Munro abscesses, parakeratosis, and acanthosis can be encountered [8].

In recent years, different imaging techniques have been studied to support the clinical diagnosis of psoriasis and to monitor and assess the therapeutic response. Cutaneous ultrasound (US) makes an important contribution to the clinical examination at the time of diagnosis, as well as in monitoring the evolution of the disease and the therapeutic response.

High-frequency ultrasound (HFUS) is a non-invasive method of morphofunctional evaluation of the structures of the epidermis, dermis, subcutaneous fat, and skin appendages. It allows high-resolution imaging and direct measurement of the thickness of various skin structures [9].

Many studies in the literature have shown that dermatoscopy is useful for assisting the non-invasive diagnosis of dermatological pathologies, including plaque psoriasis. This technique provides sub-macroscopic level information, useful in differential diagnosis [10].

Real-time imaging is made possible using confocal microscopy, which helps research the dynamic processes including inflammation and tissue remodeling in psoriatic lesions. In general, confocal microscopy can help direct therapeutic therapy options and provide insightful information about the pathophysiology of psoriasis vulgaris [11].

Treatment selection depends on factors such as the severity of the condition, the extent of skin involvement, the patient’s preferences, and any underlying medical conditions. Topical treatments, phototherapy, systemic drugs, and biological medicines that target particular immune pathways are all included in these strategies. In the era of biological therapies, numerous advancements have been achieved, yet the variability of individual responses and their monitoring still pose challenges [12].

Currently, the imaging assessment of psoriasis vulgaris plaques is a field of research in continuous evolution, with an emphasis on the development and validation of non-invasive methods for assessing the severity of skin lesions, monitoring the progress of the disease, and the response to treatment.

This article aims to illustrate the utility of imaging evaluation in assessing plaque psoriasis vulgaris, with a focus on the application of confocal microscopy and other advanced skin imaging techniques. We will examine both the benefits and constraints of these technologies, along with their potential within the framework of clinical management for psoriasis vulgaris.

## 2. Materials and Methods

We conducted a prospective study over two years (July 2022–April 2024), on patients diagnosed with moderate or severe psoriasis vulgaris, treated in the dermatology department of our institution. We selected 30 patients, of whom 15 became eligible according to the inclusion criteria (clinical lesions suggestive of psoriasis vulgaris, moderate or severe form, DLQI score ≥ 5, PASI score ≥ 5, presence of lesions on the trunk and limb) and exclusion criteria (patients who were not naive to any treatment in the last 6 months, and patients who did not come to the appointment for any reason). A total of 60 psoriasis plaques were analyzed.

The age of patients was between 26 and 77 years (mean 69.9 ± 9.4). Participants were informed in advance about the purpose and procedure of the study and provided their written consent before inclusion in the study. The study was approved by the ethics committee of the local hospital and university (ID number of the protocol: DEP153/9 May 2022) and was conducted according to the principles of the Declaration of Helsinki.

All patients were clinically consulted and the following characteristics were recorded: age, sex, lesion location, PASI score, and DLQI score, and between 2 and 6 plaques per patient were selected for paraclinical investigations. Only lesions located on the limbs and trunk were included. The selection of the two severity assessment scores was based on the hospital protocol. The dermatoscopic evaluation was performed using both a Delta 30 dermatoscope and a Vidix 4.0 videodermoscope (VD, Canfield, Italy) equipped with Vectra software (version 7.4.7) at a magnification of 20× and 70×. The images were analyzed by two experienced dermatologists who were blind to diagnosis. The analyzed criteria were chosen according to the data available in the literature, which are different for dermatoscopy and videodermoscopy [13,14].

Dermatoscopically, we analyzed the type of vascularization (dotted or linear), the vascular pattern (regular, clustered, patchy, peripheral, in rings), the distribution of the scale (patchy, peripheral, diffuse, central), the color of the scale (yellow, white), and the general erythematous background.

Through the VD, we evaluated the vascular morphology at a magnification of 70×. Globally, we divided it into two major types: twisted loops and bushy vessels. We considered bushy vessels with high condensation of loops (>3 capillary loops), medium (2–2.5 capillary loops), and low (1–1.5 capillary loops), according to Rudnicka et al. [14].

A high-resolution 20 MHz linear probe (Philips Affiniti 30) was used to perform high-frequency ultrasound. The same two skilled dermatologists carried out the examination with the probe positioned perpendicular to the plaque, no pressure applied, and with a lot of gel, until the plaque was fully entrenched. All plaques were analyzed compared to perilesional intact skin. The analyzed US characteristics were proposed by Grajdeanu et al. [9]. For each psoriasis plaque, we assessed the presence of the typical appearance in four bands with different echogenic structures. The first band (hyperechoic) being considered for the epidermis was followed by a hypoechoic band (superficial dermis with edema), a more echogenic band (deep dermis), and the last hypoechoic band (subcutaneous tissue). The thicknesses of the epidermis, the dermis of the plaque and the intact perilesional skin were measured.

The local vascularization was analyzed with color Doppler mode, adjusted to the velocities of skin circulation.

Twelve psoriasis lesions from twelve different patients were analyzed histologically and with ex vivo fluorescence confocal microscopy (FCM). Two adjacent punch biopsies were taken. One piece was analyzed histopathologically in hematoxylin-eosin. The following histological criteria were analyzed: hyperkeratosis, parakeratosis, acanthosis, elongated papillary ridges, dermal inflammatory infiltrate, presence of neutrophilic microabscesses (Munro), and hypogranularity.

The second piece was transported in 0.9% sodium chloride solution, then fixed in Tissue-Tek O.C.T and frozen at −80 degrees Celsius for 30 min. We sectioned the piece on ice with the cryotome (5 microns-thick for each section), 4 sections per slide, and 2 slides for each patient. The slides were transported in refrigerated boxes to a center specialized in confocal microscopy, to be analyzed. The analysis was performed on native sections, without using special staining.

Using a super-resolution Re-Scan Confocal Microscopy system (RCM-VIS unit) that was purchased from Confocal.nl (Amsterdam, The Netherlands), fluorescent confocal images of psoriasis plaque were captured. The system was placed on an ECLIPSE Ti2-E inverted microscope (Nikon), which had a Plan-Apochromat 10× objective (N.A. = 0.45). To be more precise, the RCM-Vis images were produced using a diode laser set at 488 nm (TOPTICA Photonics AG, Martinsried/Munich, Germany), and then the NIS Elements program (version 5.11.02) (Nikon, https://www.microscope.healthcare.nikon.com/en_EU/products/software/nis-elements, accessed on 30 January 2024) was used to analyze the data. The sections were analyzed together with a physicist with extensive experience in confocal microscopy, in order to identify the histological consecrated features correlated in fluorescence confocal microscopy. We followed the criteria proposed by Bertoni L et al. [15].

### Statistical Analysis

Statistical analysis was carried out using the MedCalc^®^ Statistical Software version 22.021 (MedCalc Software Ltd., Ostend, Belgium; https://www.medcalc.org; 2024). Data were presented as median and 25–75 percentiles or frequency and percentage, whenever appropriate. Comparisons between groups were assessed using the Mann–Whitney test.

Correlations between variables were assessed using Spearman’s rho coefficient. The Cohen kappa coefficient was used to assess the inter-method reliability. A *p*-value < 0.05 was considered statistically significant.

## 3. Results

The study included fifteen patients with psoriasis vulgaris, diagnosed histopathologically, of whom six were women and nine were men, with an average age of 55. Between two and six plaques per patient were selected and a total of sixty psoriasis plaques were analyzed by non-invasive imaging techniques. FCM analyzed twelve lesions compared to histology. Only lesions located on the limbs and trunk were included, excluding those on the scalp, genital, or acral areas. The patients underwent evaluation using two severity scores, PASI and DLQI, thereby allowing for the analysis of the quality of life among the included moderate-to-severe psoriasis patients. The main clinical characteristics evaluated are presented in Table 1. We obtained a statistically significant correlation between the PASI score and DLQI of 0.688 (*p* = 0.005).

### 3.1. High-Frequency Ultrasound

We analyzed the typical US appearance of psoriasis vulgaris in the 60 plaques. The presence of the four-band layout with a different eco structure was a constant. The hyperechoic band and the lack of damage to the subcutaneous tissue were the most common criteria. We analyzed the vascularization with the color Doppler mode; compared to the perilesional skin, it was present in 46.7% of cases (Figure 1). The main US criteria are presented in Table 2.

For each plaque, we measured the thickness of the epidermis and the dermis, compared to the intact skin. The thickness of the epidermis at the level of psoriasis plaques was greater than for the healthy skin (*p* < 0.01). The thickness of the dermis at the level of the plaques was greater than the thickness of the perilesional dermis (*p* = 0.019) (Figure 2). US thickness measurements are presented in Table 3.

### 3.2. Dermatoscopy

In most of the plaques, an erythematous background was observed, with different shades of red (which we did not quantify). Blood vessels were detected in all examined lesions, with the predominant type being dotted vessels (98.3%). The prevailing vascular pattern consisted of regularly distributed vessels (90%) (Figure 3).

We chose to analyze the distribution and color of the scales by dermatoscopy rather than VD because we could quantify the general appearance of the plaque at a lower magnification. The diffuse distribution on the entire plate (78.3%) of white scales (88.3%) was the most common pattern (Figure 4). The frequency of dermatoscopic criteria is illustrated in Table 4.

The specific aspect of psoriasis plaques localized on the limbs and trunk was a lesion with an erythematous background, with dotted vessels with regular distribution on the surface and covered by white scales with diffuse distribution.

### 3.3. Videodermatoscopy

In order to identify the vascular morphology, we performed a VD evaluation with a 70× magnification. We identified two types of vessels. The presence of bushy vessels with medium condensation was the most frequently identified pattern. Table 4 shows the frequency of VD criteria (Figure 5).

### 3.4. Histopathology and Confocal Microscopy

For 12 psoriasis plaques from 12 distinct patients, we searched for the criteria described in histopathology using confocal microscopy. The main histopathological criteria encountered are presented in Table 5.

In confocal microscopy, we considered acanthosis together with hyperkeratosis as psoriasiform hyperplasia. Good correlations were obtained between psoriasiform hyperplasia (FCM) and acanthosis (histology), and only one case of non-concordance was identified. Parakeratosis was identified in all 12 lesions, both in FCM and histologically. Neutrophilic microabscesses (Munro) were correlated with a coefficient k = 0.5 (*p* = 0.046), and FCM misdiagnosis three cases. Dermal inflammatory infiltrate was identified in all 12 lesions, both histologically and FCM. Hypogranularity had a lower correlation of k = 0.4 (*p* = 0.083) (Figure 6). Table 5 shows the frequency of FCM criteria.

## 4. Discussion

Imaging techniques play a significant role in the diagnosis and monitoring of psoriasis vulgaris, a chronic skin condition characterized by inflammation and epidermal hyperproliferation. Although the diagnosis of psoriasis is primarily based on clinical examination, imaging techniques can provide additional information and help aid in evaluating the severity and extent of lesions.

The introduction of these techniques in the evaluation of inflammatory pathologies was regarded with skepticism at the beginning, but the research of recent years shows their importance in supporting the clinical diagnosis, guiding the treatment, and monitoring the therapeutic response.

### 4.1. Cutaneous Ultrasound

Cutaneous ultrasound makes an important contribution to the clinical examination at the time of diagnosis, as well as in monitoring the evolution of the disease and the therapeutic response.

HFUS is a non-invasive method of morphofunctional evaluation of the structures of the epidermis, dermis, subcutaneous fat, and skin appendages. It allows high-resolution imaging and direct measurement of the thickness of various skin structures. Within the US analysis of the psoriasis vulgaris plaque, four bands with different echo structures are described: a hyperechoic band representing the epidermis with hyperkeratosis and parakeratosis, followed by a hypoechoic band representing the elongation of the dermal papillae, followed by a hyperechoic band corresponding to the reticular dermis, and, finally, the subcutaneous layer with the hypoechoic band. This typical aspect encountered in our study was initially presented by Grajdeanu et al. [9]. Increased thickness of the epidermis and dermis (in comparison to surrounding normal skin) is the most reliable US sign for plaque psoriasis, according to a study by Gutierrez M et al. [16]. These aspects were also encountered in our research. Also in this study, the increase in the Doppler signal, the hypoechoic band present in the upper dermis (inflammatory edema and vasodilatation in the papillary dermis), as well as the absence of subcutaneous tissue damage, were identified as signs of disease activity. Signs of therapeutic efficacy included the reduction in epidermal and dermal thickness, the disappearance of the hypoechoic band, and a decrease in the Doppler signal [16]. The increase in the thickness of the epidermis and the dermis at the level of the psoriasis plaque was a constant in our group of patients. The 20 MHz probe was found to be useful in measuring these thicknesses and in identifying dermal edema (hypoechoic band), but also in evaluating local vascularity through the color Doppler mode. In some lesions, the subtlety of the hypoechoic band made its quantification less feasible, probably requiring probes with higher frequencies (ultra-high-frequency ultrasound (UHFUS)). The disadvantage of these probes (UHFUS) is the limitation of the exploration of deep structures [17].

Recent studies in the literature show the role of cutaneous US in monitoring and evaluating the therapeutic response, with this noninvasive, repeatable imaging technique being able to identify structural changes that are not clinically apparent [18].

Lacarrubba, F. et al., in a study that included 30 patients with mild/moderate psoriasis vulgaris, demonstrated the usefulness of US (20 MHz) in evaluating the therapeutic response to clobetasol propionate 0.05% [19]. The US can identify the complications of prolonged therapy, with local cortisones such as skin atrophy [20].

The efficiency and tolerability of the keratolytic treatment with urea 50% was demonstrated by the clinical resolution of the hyperkeratosis, correlated with a 50% reduction in the thickness of the epidermis evaluated by US (22 MHz) [21]. Dini V et al. showed the correlation between the thickness of the hypoechoic band and local vascularization with the response to Ixekizumab treatment, using UHFUS (70 MHz) [22].

### 4.2. Dermatoscopy and Videodermatoscopy

Dermatoscopy and videodermatoscopy are useful, both in supporting the clinical diagnosis of psoriasis and in monitoring the therapeutic response. The most common dermatoscopic appearance is the presence of dots/red blood cells on an erythematous background observed from 97.1% to 100% in the literature [23]. In our study, the most frequent dermatoscopic appearance was an erythematous plaque, with dotted vessels with regular distribution on the surface, covered by white scales with diffuse localization. Although we only analyzed lesions from the trunk and limbs, linear vessels were identified in only one case, an aspect also encountered less frequently by Lallas et al. [13]. The change in the type and morphology of the vessels in the psoriasis plaque varies with the response to the therapy [24].

It was demonstrated that the presence of dermatoscopic hemorrhagic dots is a potent predictor for the therapeutic clinical response. Thus, a study conducted by Lallas A et al. demonstrated that the vascular pattern in psoriasis lesions correlates with the therapeutic response. Plaques with regular vascular distribution show no therapeutic response, and those with clustered vascular distribution show partial response. Therapeutic success was observed in plaques without visible vascularization [25]. In our study group, with only treatment-naive patients, we did not identify these dermatoscopic hemorrhagic dots.

VD allows the morphological and quantitative analysis of vascularization [26]. This aspect can be achieved by increasing the magnification to more than 50×. In our study, we encountered two vascular morphologies: twisted loops and bushy vessels, the most common being bushy vessels with medium condensation. In a study by Golińska J et al. on 50 patients with psoriasis vulgaris, they identified a predominance of bushy vessels on the trunk and limbs [14]. VD can also be used for the direct measurement of the vascular diameter and its variations, thus providing useful information regarding the therapeutic response [27].

### 4.3. Confocal Microscopy

In the context of psoriasis vulgaris, confocal microscopy can be used to assess structural and cellular changes characteristic of psoriatic lesions, providing detailed information on inflammation, vascularization, and cell proliferation. In recent years, a series of studies have shown the importance of using in vivo reflectance confocal microscopy (RCM) in inflammatory skin pathology, with applications in the diagnosis and monitoring of psoriasis vulgaris [11,28,29,30,31].

RCM is an imaging technique for real-time skin assessment with a resolution close to conventional histology. The contrast is obtained from the differences in the refractive index of the size of different cellular organelles and extracellular microstructures. RCM has been used to evaluate numerous inflammatory pathologies such as acute contact dermatitis and discoid cutaneous lupus, with a very good correlation with histopathology. In psoriasis vulgaris, RCM can evaluate hyperkeratosis, acanthosis, reduction or absence of the granular layer, papillomatosis, exocytosis, spongiosis, dilated blood vessels in the papillary dermis, and the presence of inflammatory cells in the papillary dermis, with a correlation of over 90% with histology [32]. RCM allows the assessment of the severity of psoriasis, at the tissue level, and consequently the choice of the optimal treatment, or the change of the current treatment. It also facilitates the evaluation of the effectiveness of the drug, the time of administration, and the choice of the dose. The main criteria in monitoring the therapeutic response are acanthosis, the number of areas with spongiosis, the number of honeycomb patterns, the number of inflammatory cells, the number of focal microabscesses, the number of non-elongated dermal papillae, and the vasculature of the dermal papillae [33].

A relatively new diagnostic technique is FCM. This involves the evaluation of tissues taken by biopsy or skin excision and subsequently marked with fluorescent dye or marker [34]. After extended research of the literature, only one study covering psoriasis plaque evaluation with FCM was found to be published. FCM showed a good correlation with histology for the identification of inflammatory infiltrate (k = 0.84), psoriasiform hyperplasia, parakeratosis, and the presence of dilated vessels [15].

In our study, we achieved satisfactory correlations with histological examination, with the strongest being observed between psoriasiform hyperplasia (FCM) and acanthosis (histology), parakeratosis, the presence of neutrophilic microabscesses (Munro), and dermal inflammatory infiltrate.

Despite the limited sample size of the twelve cases analyzed, the results are promising and indicate encouraging prospects for the future. The innovative technique proposed by us, in contrast to traditional FCM, eliminates the need for fluorescent dye but involves freezing and sectioning the tissue on ice before analysis. However, a time of at least 30 min is required for the analysis of the sections. In our study, we used FCM with utility for current research purposes, but the results are encouraging. Not all cases of psoriasis require a biopsy, the diagnosis being mainly clinical, but for difficult situations FCM could facilitate the diagnosis faster than the conventional histopathological method. Being an invasive technique that involves tissue sampling makes it more difficult for the patient to accept, this being the main disadvantage of FCM.

### 4.4. Limitations

Our study faced several limitations. We included a small number of patients, since we analyzed lesions naive to any treatment. We evaluated plaques located only on the trunk and limbs, so imaging aspects specific to the scalp, genital areas or nails were excluded. Dermatoscopy is a subjective imaging technique, dependent on the experience and expertise of the evaluator, so differences between examiners can appear and determine the decrease in diagnostic accuracy. We tried to counterbalance this aspect by parallel analysis by two dermatologists with experience in dermatoscopic diagnosis. Dermatoscopy and VD are also techniques limited to the surface of the skin, and deeper changes cannot be observed. Another limitation of dermatoscopy was represented by the areas with multiple hair follicles or plaques with thick adherent scales, making the exploration difficult to achieve.

The quality and accuracy of the US images depends on the skills and experience of the operator. The 20 MHz probe was useful in exploring the three layers of the skin, but with limitations in the detailed exploration of the dermis and epidermis. The hypoechoic band, corresponding to edema at the level of the papillary dermis, could not be quantified in all cases. US was a time-consuming technique in the evaluation of patients with lesions distributed over extensive surfaces. Artifact images were encountered in areas with multiple hair follicles or dense scales.

Despite the high-resolution images compared to histology, FCM remains an expensive and equipment-dependent method. The technique proposed by us assumed the sectioning of the specimen and therefore created the impossibility of the subsequent analysis in hematoxylin-eosin. Compared to other imaging methods, the field of view in confocal microscopy is usually smaller. This can make it difficult to analyze dynamic processes across a big area, or capture large-scale tissue architecture. FCM involves tissue harvesting, being an invasive technique, and it is therefore more difficult for patients to accept the procedure. Unlike dermatoscopy and US, FCM is not feasible for monitoring the therapeutic response.

### 4.5. Future Perspectives

Although dermatoscopy, skin ultrasound, and RCM are techniques already established in clinical evaluation and therapy monitoring in psoriasis vulgaris, future perspectives are described in the direction of artificial intelligence (AI). AI is playing an increasingly important role in medical imaging, including the diagnosis and management of psoriasis vulgaris. AI systems can be trained to recognize and classify psoriatic lesions in dermatological images, as well as in images obtained by various imaging techniques. This could facilitate the early and accurate diagnosis of psoriasis, as well as its differentiation from other skin conditions. AI can be used to identify and extract meaningful features from images, helping to develop more efficient and accurate imaging protocols for evaluating psoriasis lesions.

Currently, the histopathological examination is the gold diagnostic standard for most inflammatory dermatological pathologies. FCM shows promising results in the direction of psoriasis vulgaris, being able to be extended to broad chronic inflammatory dermatological pathologies. The short time and the less laborious technique of obtaining the images make it a serious future contender for integration into the current work protocol.

## 5. Conclusions

The role of established techniques in supporting the clinical diagnosis of psoriasis vulgaris is certain. In this study, we evaluated psoriasis plaques with different imaging techniques and proposed a new evaluation mode through FCM. The assessment and monitoring of individuals with psoriasis vulgaris can be conducted in a more complete and all-encompassing manner by incorporating dermatoscopy, ultrasonography, and confocal microscopy. By combining different imaging modalities, it may be possible to build individualized treatment plans and gain a better knowledge of the pathophysiology of psoriasis.

## Figures and Tables

**Figure 1 diagnostics-14-00969-f001:**
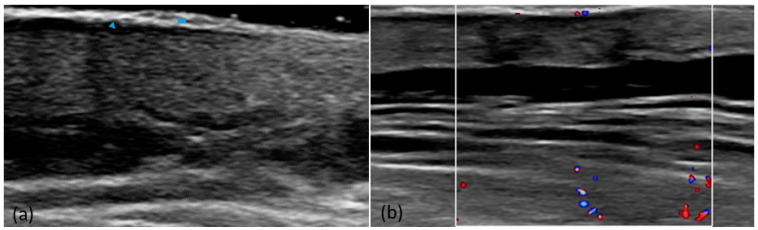
Psoriasis plaque. US shows the presence of the four-band layout with a different eco structure, with the presence of the hyperechoic band (psoriasiform hyperplasia in the epidermis) and the hypoechoic band (edema of the superficial dermis), compared to the perilesional skin (**a**). The Doppler mode shows an increase in plaque vascularity (**b**). The superficial arrow represents the hyperchoic band, the deep arrow represents the hypoechoic band.

**Figure 2 diagnostics-14-00969-f002:**
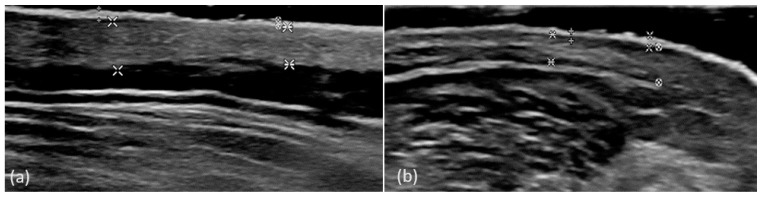
US measurements of the thickness of the epidermis (E) and the dermis (D) compared to the perilesional skin. Psoriasis plaque located in the lumbar region, E plaque = 0.108 cm, D plaque = 0.442 cm, E perilesional skin = 0.060 cm, D perilesional skin = 0.337 cm (**a**). Psoriasis plaque located in the forearm region, E plaque = 0.084 cm, D plaque = 0.241 cm, E perilesional skin = 0.066 cm, D perilesional skin = 0.187 cm (**b**).

**Figure 3 diagnostics-14-00969-f003:**
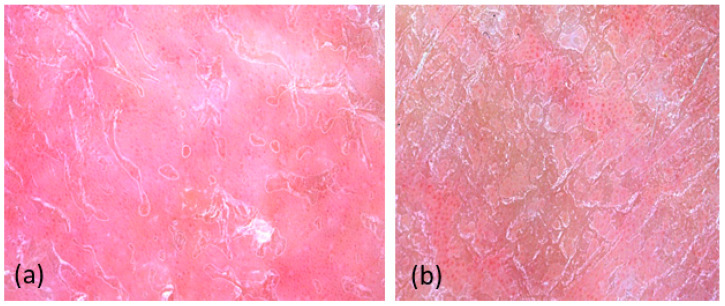
Dermoscopy shows dotted vessels on an erythematous background regularly distributed throughout the lesion (**a**). Dermoscopy shows dotted vessels in clustered distribution (**b**).

**Figure 4 diagnostics-14-00969-f004:**
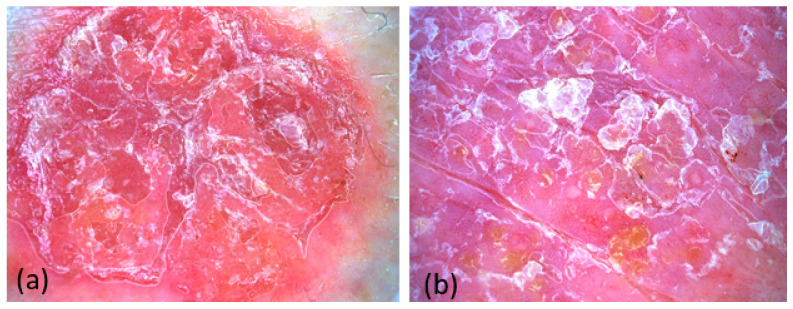
Dermatoscopy shows the presence of white scales with diffuse distribution (**a**). Dermatoscopy shows the presence of yellow scales with diffuse distribution (**b**).

**Figure 5 diagnostics-14-00969-f005:**
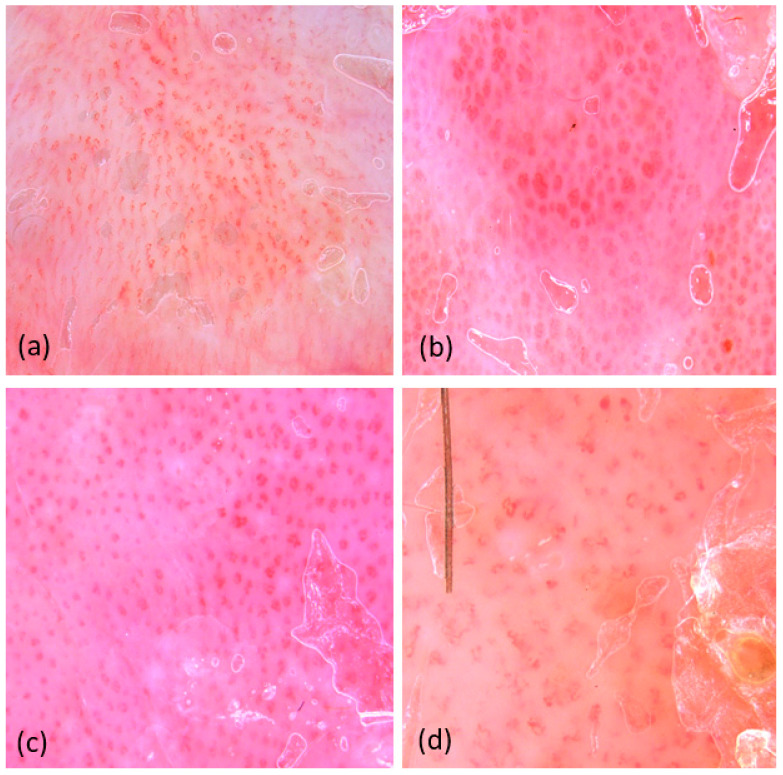
VD shows the main types of vascular morphology seen at a magnification of 70×. Twisted loops (**a**). Bushy vessels with high condensation (**b**). Bushy vessels with medium condensation (**c**). Bushy vessels with low condensation (**d**).

**Figure 6 diagnostics-14-00969-f006:**
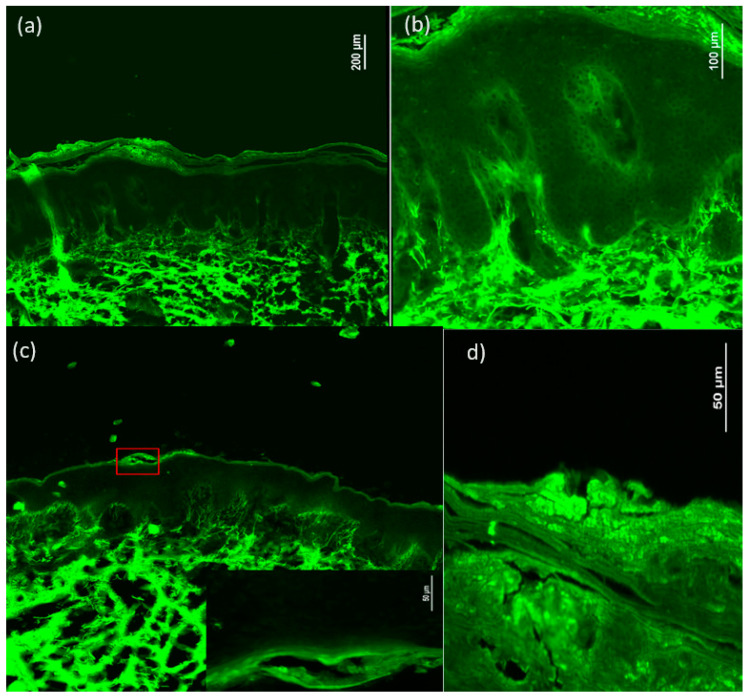
FCM of the psoriasis plaque. Overview of the psoriasis lesion; psoriasiform hyperplasia, parakeratosis, dilated capillaries in the papillary dermis, bilobate or trilobated hyperreflective nuclei, and dermal inflammatory infiltrate can be observed (**a**). Close-up image showing hypogranularity, dilated capillaries in the papillary dermis, and dermal inflammatory infiltrate (**b**). Overview image of psoriasis lesion with close-up on neutrophilic microabscesses (Munro) (**c**) Close-up image where hyperkeratosis, parakeratosis, and bilobate or trilobated hyperreflective nuclei can be observed (**d**).

**Table 1 diagnostics-14-00969-t001:** Clinical characteristics of the patients.

Variable	Characteristics		
	25th Percentile	Median	75th Percentile
Age	44	55	67
PASI score	7	9	15
DLQI score	8	10	15
No. of plaques analyzed/patient	4	4	5
Sex	female	6 (40%)	
	male	9 (60%)	
Location	limbs	39 (65%)	
	trunk	21 (35%)	

**Table 2 diagnostics-14-00969-t002:** Frequency of US psoriasis plaque characteristics.

Variable	Frequency
Hyperechoic band	60 (100%)
Hypoechoic band	48 (80%)
Lack of damage to the subcutaneous tissue	60 (100%)
Presence of vascularization	28 (46.7%)

**Table 3 diagnostics-14-00969-t003:** US thickness measurements.

Variable	Measurements		
	25th Percentile	Median	75th Percentile
Plaque epidermis thickness	0.074	0.095	0.108
Perilesional epidermis thickness	0.054	0.063	0.073
Plaque dermis thickness	0.159	0.247	0.377
Perilesional epidermis thickness	0.139	0.200	0.275

**Table 4 diagnostics-14-00969-t004:** The frequency of dermatoscopic and VD criteria for psoriasis plaque.

Dermatoscopy Criteria		Frequency
Type of vascularization	dotted	59 (98.3%)
	linear	1 (1.7%)
Vascular pattern	regular	54 (90%)
	clustered	6 (10%)
	patchy	0 (0%)
	peripheral	0 (0%)
	in rings	0 (0%)
Scale distribution	patchy	0 (0%)
	peripheral	3 (5%)
	diffuse	47 (78.3%)
	central	10 (16.7%)
Scale color	white	53 (88.3%)
	yellow	7 (11.7%)
**Vascular morphology VD**		**Frequency**
Twisted loops		8 (13.3%)
Bushy vessels	high	18 (30%)
	medium	23 (38%)
	low	11 (18.3%)

**Table 5 diagnostics-14-00969-t005:** Frequency of the main histopathological and confocal microscopy criteria.

Histopathological Criteria	Frequency
Hyperkeratosis	6 (50%)
Parakeratosis	12 (100%)
Acanthosis	11 (91.7%)
Elongated papillary ridges	8 (66.7%)
Dermal inflammatory infiltrate	12 (100%)
Presence of neutrophilic microabscesses (Munro)	9 (75%)
Hypogranularity	8 (66.7%)
**Confocal microscopy criteria**	**Frequency**
Psoriasiform hyperplasia	12 (100%)
Parakeratosis	12 (100%)
Dilated capillaries in papillary dermis	8 (66.7%)
Bilobate or trilobated hyperreflective nuclei	6 (50%)
Dermal inflammatory infiltrate	12 (100%)
Hypogranularity	4 (33.3%)

## Data Availability

The original contributions presented in the study are included in the article, further inquiries can be directed to the corresponding author.

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
