# Peer review of "Imaging Approach in the Diagnostics and Evaluation of the Psoriasis Plaque: A Preliminary Study and Literature Review"

_diagnostics, 2024, doi:10.3390/diagnostics14100969_

Round 1

Reviewer 1 Report

Comments and Suggestions for Authors

This is an interesting study regarding the findings and correlations between clinical, dermoscopic, ultrasound, confocal microscopy, and histopathological findings in plaque-type psoriasis.

The manuscript has been written well, and the findings are intriguing. While the clinical implications of imaging techniques in diagnosing and managing psoriasis are still in their early stages, the rapid progress in this field suggests that these methods will have a role to play in the clinical setting. The manuscript currently does not have a limitation section. I strongly recommend adding this section.

 Best.

Comments on the Quality of English Language

Although the English language is generally well, it is crucial to correct any grammatical errors and typos that may hinder clear communication.

Author Response

Dear Reviewer,

We thank you for taking the time to evaluate our manuscript entitled “Imaging approach in the diagnostic and evaluation of the psoriasis plaque: A preliminary study and literature review”

Below we have provided all answers and revisions in reply to all very useful suggestions.

This is an interesting study regarding the findings and correlations between clinical, dermoscopic, ultrasound, confocal microscopy, and histopathological findings in plaque-type psoriasis.

The manuscript has been written well, and the findings are intriguing. While the clinical implications of imaging techniques in diagnosing and managing psoriasis are still in their early stages, the rapid progress in this field suggests that these methods will have a role to play in the clinical setting. The manuscript currently does not have a limitation section. I strongly recommend adding this section.

Thank you for this useful suggestion. We have added the following paragraph:

"4.4. Limitations

Our study faced several limitations. We included a small number of patients, since we analyzed lesions naive to any treatment. We evaluated plaques located only at the trunk and limbs, so imaging aspects specific to the scalp, genital areas or nails were excluded. Dermatoscopy is a subjective imaging technique, dependent on the experience and expertise of the evaluator, so differences between examiners can appear and determine the decrease in diagnostic accuracy. We tried to counterbalance this aspect by parallel analysis by two dermatologists with experience in dermatoscopic diagnosis. Dermatoscopy and VD are also techniques limited to the surface of the skin, subclinical changes cannot be observed. Another limitation of dermatoscopy was represented by the areas with multiple hair follicles or plaques with thick adherent scales, making the exploration difficult to achieve.

The quality and accuracy of the US images depends on the skills and experience of the operator. The 20 MHz probe was useful in exploring the 3 layers of the skin, but with limitations in the detailed exploration of the dermis and epidermis. The hypoechoic band, corresponding to edema at the level of the papillary dermis, could not be quantified in all cases. US was a time-consuming technique in the evaluation of patients with lesions distributed over extensive surfaces. Artifact images were encountered in areas with multiple hair follicles or dense scales.

Despite the high-resolution images compared to histology, FCM remains an expensive and equipment-dependent method. The technique proposed by us, assumed the sectioning of the specimen, therefore the impossibility of the subsequent analysis in hematoxylin-eosin. Compared to other imaging methods, the field of view in confocal microscopy is usually smaller. This can make it difficult to analyze dynamic processes across a big area, or capture large-scale tissue architecture. FCM involves tissue harvesting, being an invasive technique and therefore more difficult for patients to accept the procedure. Unlike dermatoscopy and US, FCM is not feasible to monitor the therapeutic response". (line 403-430).

We have also corrected all grammatical errors and typos.

Reviewer 2 Report

Comments and Suggestions for Authors

The authors describe their experience and investigation with skin imaging in psoriasis.

1. The following statement needs a reference: "Psoriasis is considered to be the immune-mediated disease with the highest prevalence in the world (2-5% of the world's population)."

2. The use of the word "incidence" is not appropriate in the following sentence "The most common type of psoriasis encountered is plaque psoriasis with an incidence of 85-90%."

3. The introduction is relatively long and not all information is relevant to the topic.

4. To the authors' credit, there were many different visualization approaches assessed. However, 6 tables are a bit overwhelming. Would it be possible to decrease the number of tables by consolidation?

5.  This is an exploratory study and the clinical significance is completely unknown. It is highly unlikely that all of these approaches will be useful. The limitations of this study are huge and largely unrecognized by the authors.

Author Response

Dear Reviewer,

We thank you for taking the time to evaluate our manuscript entitled “Imaging approach in the diagnostic and evaluation of the psoriasis plaque: A preliminary study and literature review”

Below we have provided all answers and revisions in reply to all very useful suggestions.

  1. The following statement needs a reference: "Psoriasis is considered to be the immune-mediated disease with the highest prevalence in the world (2-5% of the world's population)."

Thank you for this useful suggestion. We added the reference [1] (Raychaudhuri, S. K.; Maverakis, E.; Raychaudhuri, S. P. Diagnosis and Classification of Psoriasis. Autoimmunity Reviews. Elsevier 2014, pp 490–495. https://doi.org/10.1016/j.autrev.2014.01.008).

  1. The use of the word "incidence" is not appropriate in the following sentence "The most common type of psoriasis encountered is plaque psoriasis with an incidence of 85-90%."

We appreciate you bringing this aspect to our attention. We replaced the word "incidence" with "proportion".

  1. The introduction is relatively long and not all information is relevant to the topic.

Thank you for this useful suggestion. We tried to condense the introduction section, we removed paragraphs (line 54-56, 58-62, 78-80).

  1. To the authors' credit, there were many different visualization approaches assessed. However, 6 tables are a bit overwhelming. Would it be possible to decrease the number of tables by consolidation?

Thank you for noticing this aspect. We condensed table 4 with 5 and 6 with 7

  1. This is an exploratory study and the clinical significance is completely unknown. It is highly unlikely that all of these approaches will be useful. The limitations of this study are huge and largely unrecognized by the authors.

We appreciate you bringing this aspect to our attention. We have added the following paragraph: "4.4. Limitations

Our study faced several limitations. We included a small number of patients, since we analyzed lesions naive to any treatment. We evaluated plaques located only at the trunk and limbs, so imaging aspects specific to the scalp, genital areas or nails were excluded. Dermatoscopy is a subjective imaging technique, dependent on the experience and expertise of the evaluator, so differences between examiners can appear and determine the decrease in diagnostic accuracy. We tried to counterbalance this aspect by parallel analysis by two dermatologists with experience in dermatoscopic diagnosis. Dermatoscopy and VD are also techniques limited to the surface of the skin, subclinical changes cannot be observed. Another limitation of dermatoscopy was represented by the areas with multiple hair follicles or plaques with thick adherent scales, making the exploration difficult to achieve.

The quality and accuracy of the US images depends on the skills and experience of the operator. The 20 MHz probe was useful in exploring the 3 layers of the skin, but with limitations in the detailed exploration of the dermis and epidermis. The hypoechoic band, corresponding to edema at the level of the papillary dermis, could not be quantified in all cases. US was a time-consuming technique in the evaluation of patients with lesions distributed over extensive surfaces. Artifact images were encountered in areas with multiple hair follicles or dense scales.

Despite the high-resolution images compared to histology, FCM remains an expensive and equipment-dependent method. The technique proposed by us, assumed the sectioning of the specimen, therefore the impossibility of the subsequent analysis in hematoxylin-eosin. Compared to other imaging methods, the field of view in confocal microscopy is usually smaller. This can make it difficult to analyze dynamic processes across a big area, or capture large-scale tissue architecture. FCM involves tissue harvesting, being an invasive technique and therefore more difficult for patients to accept the procedure. Unlike dermatoscopy and US, FCM is not feasible to monitor the therapeutic response". (line 403-430).